# Pre-Conception Maternal Food Intake and the Association with Childhood Allergies

**DOI:** 10.3390/nu11081851

**Published:** 2019-08-09

**Authors:** Jessica A. Grieger, Anita M. Pelecanos, Cameron Hurst, Andrew Tai, Vicki L. Clifton

**Affiliations:** 1Robinson Research Institute, University of Adelaide, North Adelaide, South Australia 5000, Australia; 2Discipline of Obstetrics and Gynaecology, Adelaide Medical School, University of Adelaide, Adelaide, South Australia 5000, Australia; 3Statistics Unit, QIMR Berghofer Medical Research Institute, Brisbane 4006, Australia; 4Department of Respiratory and Sleep medicine, Women’s and Children’s Hospital, Adelaide 5006, Australia; 5Mater Medical Research Institute, University of Queensland, Brisbane 4101, Australia

**Keywords:** childhood allergy, eczema, wheeze, rhinitis, developmental programming, food intake, pre-conception diet, pregnancy

## Abstract

Background: Periconceptional nutrition may have an important function in programming the immune function and allergies, however, there is a lack of studies assessing pre-conception food intake and childhood allergic disorders. The aim of the current study was to identify maternal pre-conception dietary components that may be associated with allergic disorders in children up to 3 years of age. Methods: Pregnant women attending their first antenatal visit and who were aged >18 years were invited to participate. Pre-conception food frequency data was retrospectively collected at 18 weeks’ gestation. Childhood eczema, current wheeze, and rhinitis was assessed at 36 months of age using a questionnaire and doctor diagnosis (*n* = 234). Linear discriminant analysis (LDA) was used to explore the combination of dietary food components that best discriminated between allergy status in children. Results: Maternal pre-conception food intake such as low and high fat dairy, fresh fruit, unsaturated spreads, and take-away foods, were protective for any allergy assessed. Non-oily fish was protective for eczema and current wheeze; saturated spreads (e.g., butter) was protective for eczema, current wheeze, and rhinitis; poultry and fruit juice were adversely associated with each allergy. Conclusions: Pre-conception food intakes demonstrate inconsistent and somewhat contrary relationships to the development of child allergies. Whether and how maternal food intake impacts the underlying fetal programming and the mechanisms of childhood allergy warrants further investigation.

## 1. Introduction

The prevalence of asthma and allergies among children has become an increasing problem in the last few decades. The International Study of Asthma and Allergies in Childhood (ISAAC) has consistently identified Australia, along with the UK, New Zealand and the Republic of Ireland, as having a relatively high prevalence of asthma in children [1,2]. In Australia, the incidence of asthma or wheeze in children aged 2–3 years was 6.4% and 15.4%, respectively [3]. In Western countries, the prevalence of food allergies in children is approximately 4–7% but in up to 10% of infants has been reported in Australia [4]. Moreover, in regions of social disadvantage, the prevalence of asthma [5], rhinitis, and sensitisation is higher [6], highlighting the importance of identifying modifiable factors that may contribute to allergy development in children.

Extensive evidence implicates that exposures and events during the critical stages in pregnancy can alter offspring phenotype and disease predisposition in later life. In particular, periconceptional nutrition may have an important function in programming immune function and allergies. There is evidence to suggest that higher intakes of oily fish, omega 3 fatty acid, or high dose omega 3 supplements in early or late pregnancy may reduce the risk of offspring atopic disease [7,8,9,10,11,12,13], and higher total and individual dairy products were associated with a reduction in childhood allergic disorders [14,15,16]. The results were inconsistent for fruit or vegetable consumption, showing either a reduced risk for asthma or wheeze [13], no association [12], or an increased risk of sensitization against food allergens at 2 years [10]. In a recent systematic review of observational and intervention studies, there were no consistent associations for a range of maternal dietary exposures including fruits, vegetables, citrus fruits, nuts, cereal, milk, or egg intake in pregnancy, and the risk of allergic outcomes during the first year of life [17]. The analysis however showed a high risk of bias, the search strategy was only up until 2013, neglecting a series of studies post this date, and the identification of allergies in the first year of life may not represent the same prevalence identified in slightly older children.

In regions of low socioeconomic status, diet quality is poor [18,19], and both low socioeconomic status [6] and poor diet quality [11,20] associates with a higher risk of allergies. The authors have previously shown that a pre-conception high fat/sugar/takeaway dietary pattern was associated with an increased likelihood of uncontrolled asthma in pregnancy [21]. However, while there is the notion that maternal diet during pregnancy may play a role in infant allergy outcomes, there is a lack of studies assessing pre-conception food intake and childhood allergic disorders. The aim of the current study was to identify maternal pre-conception nutrition that may be associated with allergic disorders in children up to 3 years of age.

## 2. Materials and Methods

### 2.1. Study Setting and Population

This study is a secondary analysis of a larger prospective cohort study from the Lyell McEwin Hospital, Adelaide, South Australia, Australia, that assessed the effects of asthma during pregnancy on the mother, placenta, and baby [22]. The Lyell McEwin Hospital is a tertiary teaching hospital in a socially disadvantaged area of northern Adelaide. Pregnant women attending their first antenatal visit and who were aged >18 years were invited to participate. The recruitment and data collection for all women and their babies took place between May 2009 and July 2013. Of the 400 pregnant women who consented, 91 women withdrew from additional participation because of the following: (1) Unable to be contacted (*n* = 10); (2) could not attend appointments/not interested (*n* = 56); (3) relocated (*n* = 9); (4) miscarriage (*n* = 12); or (5) voluntary termination of the pregnancy (*n* = 4). This left 309 women included in the study. The project was approved by The Queen Elizabeth Hospital and Lyell McEwin Hospital Human Research Ethics Committee and The University of Adelaide Human Research Ethics Committee. All women gave written informed consent.

### 2.2. Maternal Data Collection

At the first antenatal clinic visit, at a median 13 weeks’ gestation, the demographics were collected. These included maternal age and ethnicity, body weight measured to the nearest 0.1 kg using calibrated electronic scales (Professional Medical Scale, ScalesPlus, Australia) and height was measured by a wall-mounted stadiometer. BMI was calculated as (weight (kg))/(height (m))^2^ and used as a proxy measure of usual weight status. Maternal asthma status was determined by the response to the question by the midwife, ”Have you been told by a doctor that you have asthma?” and ”Have you used any asthma medications in the last year like salbutamol or a preventer?”. The women were categorised as non-asthmatic or asthmatic. Asthma exacerbations were determined using questions asked by the midwife, as previously reported and categorised as controlled, mild exacerbation, or severe exacerbations [21]. Obstetric history, medical, mental and surgical health information, and socioeconomic status (Socio-Economic Indexes for Areas) were obtained from medical records. Gestational age was determined by the date of the last menstrual period and confirmed at the 18-week ultrasound. Smoking history was recorded and women were categorised as non-smoker/former smoker (the median duration that former smokers had last smoked was a median of 3 years prior to pregnancy)), women who quit smoking during pregnancy (median time was at 5 weeks’ gestation) or current smokers (continuing to smoke in pregnancy).

### 2.3. Maternal Dietary Intake

At 18 weeks’ gestation, the validated Cancer Council of Victoria’s Dietary Questionnaire for Epidemiological Studies Food Frequency Questionnaire (FFQ) was used to obtain dietary intake data covering the 12 months before pregnancy (http://www.cancervic.org.au/downloads/cec/FFQs/FFQ_sample_watermark.pdf). One hundred different foods (grams per day) were obtained from the FFQ and were assigned into 33 food components (grams per day) based on a previous Australian study [23] and used for analysis.

### 2.4. Child Allergy Outcomes

The follow-up data was collected on infants every 12 months from 6 months of age until 36 months of age, as described in our previous paper [22], but this study only reported data at 36 months of age. Briefly, allergy (eczema, current wheeze, and rhinitis) was determined based on the child’s history and clinical tests by their general practitioner and/or allergy specialist and reported to the parent, who then completed a modified version of the International Study of Asthma and Allergy in Childhood (ISAAC) questionnaire [24]. Key questions from the ISAAC questionnaire were used to gather the data on the symptoms of atopic eczema, current wheeze, and allergic rhinitis. Eczema was defined if the parents reported “Yes” to any one of the following questions: “In the last 12 months, has your child had a dry itchy rash at any time?” and “Has your child ever had eczema?” The questions were asked for both current wheeze and asthma using the following questions: “Has your child ever had wheezing or whistling in the chest at any time in the past?” (yes/no); “Has your child had wheezing or whistling in the chest in the last 12 months?” (yes/no); “Has your doctor ever told you that your child has asthma?” The GP asked about a history of recurrent wheeze and/or cough responding to the bronchodilator treatment, and although the asthma diagnosis is based on clinical grounds, there is ambiguity regarding asthma diagnosis in young children. This study labeled this outcome as current wheeze for any yes response to these questions. Rhinitis was defined if the parents reported “Yes” to the question, “Has your child ever had hayfever?”, or if parents responded yes to both questions, “In the last 12 months, has your child had a problem with sneezing, or a runny, or a blocked nose when he/she did not have a cold or the flu?” and “In the last 12 months, has this nose problem been accompanied by itchy/watery eyes?” The primary outcome was an allergy status in children. It was defined as the presence of eczema, current wheeze, and/or rhinitis. The three secondary outcomes were the presence of eczema, current wheeze, or rhinitis. If a child had more than one allergy for the secondary outcomes, they were classified as positive for the allergy in question.

### 2.5. Statistical Methods

The summary statistics in the form of the number (percent), the mean (standard deviation) or median (interquartile range) were reported. The dietary food components were scaled and a linear discriminant analysis was used to explore the combination of dietary food components that best discriminated between allergy statuses in children. The cases with incomplete data were excluded for analysis (complete case analysis). The linear discriminant analysis was performed on all 33 dietary food components. To gauge the predictive accuracies of the models, both model sensitivity and specificity was reported. The analysis was performed in R (v3.5.2; R Core team, 2018, Vienna, Austria) and the linear discriminant analysis was performed using the R package MASS [25].

## 3. Results

Of the 309 women who completed the pre-conception FFQ, 273 had consented for follow up of childhood allergy outcomes and 234 had completed the pre-conception diet data. The maternal and neonatal characteristics of those with completed the dietary data are reported in Table 1. The median BMI was in the overweight range, and more than one third were obese (Table 1). Over half of the study group were in the lowest socioeconomic status group, with 75% being former or non-smokers. Approximately half of the women had asthma, of whom half had controlled asthma and half had mild or severe exacerbations.

### 3.1. Food Components and Their Association with Allergy

Figure 1 outlines the linear discriminant coefficients (LDCs) of the food components with allergy status. The positive coefficients indicate the food component is positively associated with allergy presence, while the negative coefficients indicate the food component is protective. Various foods within the same food group, for example, tomatoes, red/yellow vegetables, leafy green vegetables, were not consistently positively or negatively related to allergy status in children. Poultry and fruit juice (i.e., 100% fruit juice) both had a positive relationship with children having the allergies of eczema, current wheeze, and rhinitis, while saturated spreads such as butter, refined grains and takeaway were all negatively related with these allergies. A high negative association with current wheeze was observed for low fat dairy (LDC −0.78). However, low fat dairy had a positive association with rhinitis (LDC 0.31) and eczema (LDC 0.25). Other fish (e.g., canned fish and non-fried fish) were positively related to rhinitis (LDC 0.28) but negatively related to eczema (LDC −0.32) and current wheeze (LDC −0.31).

Higher amounts of leafy green vegetables and potatoes coincided with rhinitis development, showing high LDCs of 0.58 and 0.47 respectively. Canned fruit had a highly negative association with rhinitis development (LDC −0.51). Refined grains had a highly negative association with eczema development (LDC −0.51). Salty type foods including vegemite (a yeast extract spread, similar to Marmite), tomato sauce, and processed meats, were associated with the development of eczema.

### 3.2. Prediction of Food Components and Allergy Status

The prediction accuracies of the linear discriminant analysis (LDA) food component models predicting the allergy status were poor to moderate (Table 2). For any allergy, most were classified as having an allergy, resulting in very high sensitivity of 97.9% but a very poor specificity of 23.9%. Although a high specificity was observed for rhinitis (85.8%), the sensitivity was poor (32.3%). Current wheeze and eczema had low to moderate accuracy measures.

## 4. Discussion

This study adds a unique contribution to the literature assessing pre-conception dietary components and their association with the development of allergies in children up to 3 years of age. In this group of pregnant women of relatively low socioeconomic status, of whom 60% were either overweight or obese (measured at the end of the first trimester), and approximately 50% were asthmatic, maternal pre-conception intake of a range of core and discretionary food choices appeared protective of “any childhood allergy”. However, for individual allergies, refined grains, takeaway foods and saturated spreads appeared protective of childhood eczema, wheeze, and rhinitis; wholegrains, other vegetables, and tomatoes were protective of rhinitis; and fish (i.e., canned fish and non-fried fish) was protective for wheeze and eczema. Given the persistent increase in allergic disorders [1] and the global deterioration in diet quality [26], identifying key foods that may contribute to the development of child allergies is of clear importance.

The level of social deprivation in this pregnant population is significant. Data from the Lyell McEwin Hospital service has previously demonstrated that 36% of pregnant women reported they were abused as children, 35% reported major life stressors and 30% reported a diagnosis of depression during antenatal care [27]. Population-based data also reported that 40% of individuals in this community did not finish year 10 at high school, more than 20% were unemployed, 27% were housed by the government and 22% of families only had a female parent [28]. Not surprisingly, the diet quality in the areas of low socioeconomic status is poor [29]. Additionally, more than a third of these women were obese, which is associated with low socioeconomic status [30], poor diet quality [31] and the development of atopy [32]. The burden of environmental risk factors is impacted by socioeconomic position, highlighting the need to address such modifiable behaviours, particularly before and during pregnancy, to optimize the health of their offspring.

Overall, the literature on maternal food intake and dietary patterns assessed during pregnancy and the development of childhood allergies are inconsistent. Typically, consumption of fish or omega 3 polyunsaturated fatty acids appears to be protective [7,8,9,10,12,13], with other reported food intakes showing null, positive, or adverse outcomes [10,12,13,14,15,16,17]. Moreover, there is uncertainty as to what time point during pregnancy, or whether before or straight after pregnancy, may have the biggest influence on allergy outcomes [17]. There is some data to suggest that periconceptional nutrition may have an important function in programming immune function and allergies, in particular, known immune-modulatory agents including long-chain omega-3 polyunsaturated fatty acids [11], antioxidant vitamins, vitamin D and folate [33], but the precise mechanisms operating before and during pregnancy that link maternal nutrition to offspring allergy development have not been clearly established.

This study found that the consumption of fish (i.e., canned fish and non-fried fish) may be protective for the development of child wheeze and eczema, but was adversely associated with rhinitis. Only one recently published study was found assessing pre-pregnancy diet on allergies in children, finding no association between maternal fish intake in the 1 year prior to pregnancy and any type of child allergy [34]. In a meta-analysis of maternal diet during pregnancy and infancy and risk of allergic or autoimmune disease, fish oil supplementation during pregnancy and breastfeeding was associated with a reduced risk of sensitization to food allergens but inconsistent results were found for the association with other allergic outcomes [17]. Observational studies however were inconclusive regarding the consumption of fish or fatty acids on allergy outcomes [17]. While individual studies demonstrated a protective effect of maternal fish and omega 3 fatty acids intake on offspring allergies, the synthesized effect has been less convincing. Further mechanistic studies are needed to identify the best time point for consuming fish and omega 3 fatty supplements, and the optimal intake to reduce the development of allergies in offspring.

The protective effect of low fat dairy consumption on the risk of wheeze corroborated the findings by Hallit et al. during pregnancy [35], but we report an adverse effect for eczema and rhinitis in contrast to other findings for eczema [15,36]. Comparatively, Maslova et al. reported a mid-pregnancy intake of low-fat yoghurt increased the risk of child asthma and allergic rhinitis at 7 years, while whole milk was protective at 18 months of age [37]. The Differences in the results may be due to the nutrient profile of dairy products, which include saturated fatty acids, magnesium, zinc, and calcium, whereas yogurt additionally contains probiotics, and sometimes artificial sweeteners [38]. Whilst the role of additives in relation to the inflammation and allergies is unclear [39,40], it may be that the specific type of dairy product consumed is more important in relation to allergy outcomes. Moreover, the assessment of asthma is difficult in 3 years old, thus comparing our outcome of current wheeze in 3 years old to asthma in 7 years old may reflect a different aetiology.

The consumption of fruit juice was adversely associated with each of the allergies assessed, whereas fresh fruit was protective for any allergy, and canned fruit protective for rhinitis development. While these specific results support recommendations to consume fresh fruit and limit the consumption of fruit juice [41], there were inconsistent associations with fruit intake for other allergic outcomes, which are also reflected in the literature. Leafy green vegetables unexpectedly showed a positive association with rhinitis. The context in which these foods were eaten were not investigated, for example, leafy vegetables may have been consumed in a burger or spinach on a pizza, so it is difficult to interpret such findings.

This study further found that higher pre-pregnancy intakes of refined grains, take-away, and saturated spreads were protective of childhood allergies. Two studies found that frequent fast food consumption during pregnancy was associated with asthmatic symptoms in young children [42] and adolescents [43]. However, two other studies reported maternal intake of foods high in saturated fats was associated with a decrease in upper airway symptoms at age 9–11 years [44] and a high intake of total saturated fatty acids was associated with a decreased risk of asthma at 5 years [45]. While there are mechanisms through which dietary lipids exert pro-inflammatory or anti-inflammatory functions on the cells of the innate immune system [46] and airway responses [47], the challenge is in identifying how and by what mechanisms maternal intakes prior to pregnancy may affect allergy development in children. Furthermore, while several agencies recommend to limit the intake of saturated fats to support cardiovascular health in adults [48,49,50], such recommendations may not be appropriate for pregnancy and in particular, the development of childhood allergies.

Some of our results may be due to reverse causation. While it may be expected that women with a family history of allergies have poor quality diets, contributing to the allergy in the child, this study found that some healthier foods were associated with child allergies. This may be indicative of a behaviour change bias, where individuals adopt more positive behaviours over time or even exclude certain foods based on an allergy diagnosis. As the newly adopted dietary intake is assessed, this systematic misclassification may result in an apparent association of a healthier diet with the development of child allergies, which could readily be interpreted as a protective effect of a poor-quality diet. While it is not possible to identify, test and model out this bias given the limitations inherent in our study design, if present, such systematic misclassification would be expected to bias risk estimates towards the null.

There were some limitations in our study. First, the results may be influenced by selection bias: The study population was from a socially disadvantaged area, one third of the women were overweight/obese, and half were asthmatic. All of these factors have been previously associated with the development of allergies in offspring [6,32,51,52]. Thus, the observed effect this study reports may be exaggerated and it is not possible to assume an effect in women without these characteristics or who were not included in the study. Nevertheless, this study adds to the maternal diet and allergy literature, which has generally shown inconsistent relationships. Second, the FFQ was used to measure diet among the study participants. Although the FFQ is widely used in dietary studies and has been validated in several populations [53,54,55], there are a number of biases that can be easily introduced into measuring diet including those associated with participant recall, compliance, and even the intake of allergenic foods in pregnancy [56]. Recall bias may have been particularly problematic since the participants were asked to recount their pre-conception diet only after they had conceived. Third, this study is limited regarding the statements about the causal link between pre-conception diet and allergy status due to the observational nature of this study. For example, diet quality has consistently been associated with maternal BMI and smoking status, which are driven by socio-demographic factors. Untangling the complex interplay between these factors was something not within the scope of the present study. Assessment of medication use (except required for asthma) or supplement use, was not thoroughly collected in this dataset, but it is acknowledged that their use may be associated with some of our findings (prevalence of allergies or association with dietary intake). Fourth, the LDA analysis is a true multivariate method and does not really lend itself to confounding bias adjustments that are seen in the classical biostatistical models such as logistic regression. Thus, while LDA accounts for correlations among the individual dietary items when classifying allergy status, the impact of other maternal exposures such as body weight, medication and supplement use, smoking, breastfeeding, and even hygiene practices, cannot be fully elucidated. Finally, the relatively modest sample size meant the authors were unable to cross-validate the model, so the generalizability of these findings to the population is uncertain. Consequently, our findings need to be recognised in terms of an exploratory analysis, in that this study had interesting observed associations that warrant further investigation.

This study did have some strengths. To the authors’ knowledge, this is the first study that has applied LDA to examine the association of pre-conception dietary food groups with child allergy outcomes. A linear discriminant analysis has the advantage of accounting for how food groups vary together, as opposed to examining elements individually, and only then, adding them to the model. The LDAs multivariate nature should adequately reflect the inter-relatedness of food components within a person’s diet. This study also used a well characterised pregnant population with knowledge of maternal asthma and allergies, BMI, medications, mental health, smoking status as well as diet which all could impact on the future risk of child allergy development.

## 5. Conclusions

Preconception food intakes demonstrate inconsistent and somewhat contrary relationships toward the development of child allergies. This study builds on the single previous study that has so far assessed maternal pre-pregnancy dietary intakes and describe some consistent relationships to those assessing maternal food intake during pregnancy. While it is established that maternal food intake plays a key role in fetal growth and development, whether and how maternal food intake impacts the underlying fetal programming and mechanisms of childhood allergies warrants further investigation.

## Figures and Tables

**Figure 1 nutrients-11-01851-f001:**
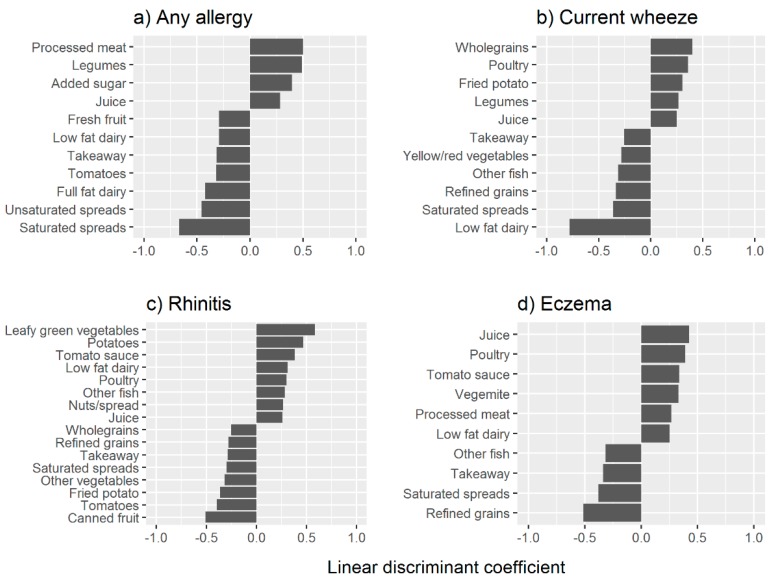
The linear discriminant coefficients >0.25 of dietary components of (**a**) any allergy, (**b**) current wheeze, (**c**) rhinitis and (**d**) eczema. The coefficients <−0.25 suggest a protective effect and coefficients >0.25 indicate a risky effect for the allergy status. The coefficients between −0.25 and 0.25 are not presented.

**Table 1 nutrients-11-01851-t001:** Maternal and neonatal characteristics.

Maternal	Total (*n* = 234)
Age (years), mean (SD)	26.9 (5.6)
Weight (kg), median (IQR)	73 (61–85)
Body mass index (kg/m^2^), median (IQR)	27.4 (23.2–32.0)
Body mass index category, *n* (%)	
<25 kg/m^2^	94 (40.3%)
25–29 kg/m^2^	57 (24.5%)
≥30 kg/m^2^	82 (35.2%)
Socioeconomic status, *n* (%)	
5 (highest)	11 (4.7%)
4	18 (7.7%)
3	8 (3.4%)
2	66 (28.2%)
1 (lowest)	131 (56.0%)
Ethnicity, *n* (%)	
Caucasian	217 (92.7%)
Non-Caucasian	17 (7.3%)
Smoking status, *n* (%)	
Non-smoker/former smoker	175 (74.8%)
Quit during pregnancy	24 (10.3%)
Current smoker	35 (15.0%)
Asthma status, *n* (%)	
Non-asthmatic	108 (46.2%)
Asthmatic	126 (53.8%)
Asthma exacerbations, *n* (%)	
Non-asthmatic	108 (46.2%)
Controlled	61 (26.1%)
Mild exacerbation	33 (14.1%)
Severe exacerbation	32 (13.7%)
Gravida, *n* (%)	
0–1	66 (28.4%)
≥2	166 (71.6%)
Parity, *n* (%)	
0	92 (39.8%)
≥1	139 (60.2%)
Neonatal	
Birthweight (g), mean (SD)	3405 (564)
Length (cm), mean (SD) ^1^	49.8 (3.0)
Head Circumference (cm), median (IQR) ^1^	35 (34–36)
Sex, *n* (%)	
Male	117 (50%)
Female	117 (50%)
Gestation, weeks, median (IQR)	39 (38–40)
Allergy status at 3 years, *n* (%)	
Any allergy/No allergy	188 (80%)/46 (20%)
Eczema/No eczema	105 (45%)/129 (55%)
Current wheeze/No current wheeze	131 (56%)/103 (44%)
Rhinitis/No rhinitis	93 (40%)/141 (60%)

^1^*n* = 214 (assessment not taken).

**Table 2 nutrients-11-01851-t002:** Prediction accuracy of LDA (linear discriminant analysis).

Predicted Allergy Status	Actual Allergy Status	Sensitivity	Specificity
Absent	Present
Any allergy				
Absent	11	4	97.9%	23.9%
Present	35	184		
Eczema				
Absent	99	53	49.5%	76.7%
Present	30	52		
Current wheeze				
Absent	55	32	75.6%	53.4%
Present	48	99		
Rhinitis				
Absent	121	63	32.3%	85.8%
Present	20	30

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
