# Peer review of "Pre-Conception Maternal Food Intake and the Association with Childhood Allergies"

_nutrients, 2019, doi:10.3390/nu11081851_

Round 1

Reviewer 1 Report

Review Nutrients nutrients-560064 (2019)

Title: Periconceptional maternal food intake and the association with childhood allergies.

The manuscript deals with the diet/nutrition around conception and start of pregnancy, and the influence on the sensitization and/or allergy status of the children in the first 3 years of life.

The study question is very interesting and important to the field. The content is described clearly, the length and references are appropriate.

The study includes a valuable data set and a reasonable sample size for a basic study to start investigating influence factors for allergy development exactly within such a special group, i.e. here with a high percentage of low SES, many affected by asthma (53%) and high BMI. Probably, these special characteristics should also be mentioned in the Conclusion (already in Abstract).

The duration of observation for up to years of age of the children is positive, clearly it is worth following them up until after school age, where it is known that many allergies are outgrown.

Major comments:

·      As stated in the discussion, there are a lot of factors possibly contributing to the outcome, which could not be detangled here:

o   Obesity -present in one third of this study population- may contribute per se; please discuss and cite literature

o   Indirectly, these women may take medication associated with their BMI, e.g. acid-suppressing medication due to reflux and heartburn – this medication per se increases the risk of allergies; please discuss and cite literature

·      Why was “food allergy” not considered as outcome measure? (line 101)

·      Was breast-feeding taken into consideration or even asked for in the questionnaire?

·      Even though it was not the major outcome focus, the numbers of sensitized/allergic/affected children among the neonates and over the 3-year-observatioin period should be given (actual numbers or as percentages)

·      In Table 1, besides asthma status of mothers, also their allergy status should be reported

·      In the discussion, it needs to be stressed in more detail that a lot of factors have not been considered in the (statistical) evaluation, like medications for reflux, stress, depression…….; breast-feeding; smoking of mother and passive smoking for neonate/child; hygiene of child (bathing, moisturizers etc.)

Minor comments:

·      Sentence in Conclusion (also in Abstract) is unclear: “inconsistent” and “contrary” to what? Existing studies? Within the study? Inconsistent from one type of allergy to the other?

·      Introduction/line40: incidence of food allergies in Australia also very high!??!? Please state and give references

·      “Former smokers”: for how long at minimum have these individuals been non-smokers

·      Saturated spread: give example

·      Include abbreviation list or give full name when first mentioned (e.g. for SES)

·      Line 69: give citation if this first prospective study is already published

·      Line 91: and women who proceed with smoking during their pregnancy?! (acc. to table 1, this would be 15%)

·      Line 94: if there is access, a link to the full FFQ would be good to include in the paper

·      Line 101: “was determined by their general practitioner” – how? By assessing history and examining symptoms, or doing tests?

·      Table 1: “Asthma status” – was this assessed at gestational week 18? Did those women have asthma also already before conception?

·      Table 1: for higher clarity, please list “non asthmatic” before “asthmatic” and not before “asthma exacerbations” (like listed in line 85/86)

·      Table 1: please add info why only n=214 for head circumference – data for others not available (not measured or info not given by mother?

·      Line 129: “shared a common food group” is not clear;

Line 129: “meats/legumes” as a common food group?

·      Line 135: “Other fish” – explain or give example

·      Fig. 1: what is Vegemite?

·      Please define “juice”

·      Line 173: again, define “fish” clearly or give example

·      Line 204: typo “and a and high”

Author Response

REVIEWER 1

The manuscript deals with the diet/nutrition around conception and start of pregnancy, and the influence on the sensitization and/or allergy status of the children in the first 3 years of life. The study question is very interesting and important to the field. The content is described clearly, the length and references are appropriate.

The study includes a valuable data set and a reasonable sample size for a basic study to start investigating influence factors for allergy development exactly within such a special group, i.e. here with a high percentage of low SES, many affected by asthma (53%) and high BMI. Probably, these special characteristics should also be mentioned in the Conclusion (already in Abstract).

The duration of observation for up to years of age of the children is positive, clearly it is worth following them up until after school age, where it is known that many allergies are outgrown.

Major comments:

As stated in the discussion, there are a lot of factors possibly contributing to the outcome, which could not be detangled here:

Comment 1: Obesity -present in one third of this study population- may contribute per se; please discuss and cite literature:

Response 1: Thankyou for this comment. We now include in the discussion (page 7):

“Additionally, more than a third of these women were obese, which is associated with low socioeconomic status (Gomes D 2019), poor diet quality (Asghari G 2017) and the development of atopy (Boulet L-P 2015). The burden of environmental risk factors is impacted by socioeconomic position, highlighting the need to address such modifiable behaviours, particularly before and during pregnancy, to optimise the health of their offspring.”

Comment 2:Indirectly, these women may take medication associated with their BMI, e.g. acid-suppressing medication due to reflux and heartburn – this medication per se increases the risk of allergies; please discuss and cite literature

Response 2: Unfortunately we were not able to assess complete medication use in this cohort, and we have acknowledged this as a limitation in the discussion (page 8):

“Moreover, assessment of medication use (except required for asthma) or supplement use, was not thoroughly collected in this dataset, but we acknowledge their use may be associated with some of our findings (prevalence of allergies or association with dietary intake)”.

Comment 3:Why was “food allergy” not considered as outcome measure? (line 101)

Response 3: Food allergy was not collected using the ISAAC questionnaire and we did not have an appropriate protocol to collect this information. We have removed the mention of ‘food allergy’ from the methods.

Comment 4: Was breast-feeding taken into consideration or even asked for in the questionnaire?

Response 4: Thankyou for this comment. With the approach we made for the LDA analysis, including confounding variables such as breastfeeding, would have made interpretation of the results difficult. We understand that this is a limitation in our approach, but is appropriate given the exploratory nature of the study. As also specified in response 7 below, we have added the following text (page 8 discussion): 

“The LDA analysis is a true multivariate method and does not really lend itself to confounding bias adjustment that we see in classical biostatistical models such as logistic regression. Thus, while LDA accounts for correlations among the individual dietary items when classifying allergy status, the impact of other maternal exposures such as body weight, medication and supplement use, smoking, breastfeeding, and even hygiene practices, cannot be fully elucidated.”

Comment 5:Even though it was not the major outcome focus, the numbers of sensitized/allergic/affected children among the neonates and over the 3-year-observatioin period should be given (actual numbers or as percentages)

Response 5: We have included at the bottom of Table 1:

Allergy status at 3 years, n (%)

Any allergy / No allergy188 (80%) / 46 (20%)

Asthma / No asthma131 (56%) / 103 (44%)

Rhinitis / No rhinitis93 (40%) / 141 (60%)

Eczema / No eczema105 (45%) / 129 (55%)

Comment 6:In Table 1, besides asthma status of mothers, also their allergy status should be reported.

Response 6: Because the original cohort was based on asthma and pregnancy outcomes, unfortunately we do not have information on other allergies in these women.

Comment 7: In the discussion, it needs to be stressed in more detail that a lot of factors have not been considered in the (statistical) evaluation, like medications for reflux, stress, depression…….; breast-feeding; smoking of mother and passive smoking for neonate/child; hygiene of child (bathing, moisturizers etc.)

Response 7:  Thank you for this comment. This is indeed a limitation of LDA and it was an issue that came up in our analytical planning. Initially we considered binary logistic regression which would allow both dietary and other potential confounders to be included into a prediction model of allergy status. However, such a multivariable logistic regression model is not truly multivariate. Logistic regression would account for the variation in individual dietary items, but not their covariation (correlation).  An alternative approach would have been to include confounders into the LDA along with the dietary items, but we opted not to do this not only because it would have made interpretation of the results difficult, but this would be unlikely to have (at least fully) solved the problem of confounding. Having said this, we do appreciate that this is a limitation in our approach, but is appropriate given the exploratory nature of the study. We have now added the following text (page 8 discussion): 

“The LDA analysis is a true multivariate method and does not really lend itself to confounding bias adjustment that we see in classical biostatistical models such as logistic regression. Thus, while LDA accounts for correlations among the individual dietary items when classifying allergy status, the impact of other maternal exposures such as body weight, medication and supplement use, smoking, breastfeeding, and even hygiene practices, cannot be fully elucidated.”

Minor comments:

Comment 8:Sentence in Conclusion (also in Abstract) is unclear: “inconsistent” and “contrary” to what? Existing studies? Within the study? Inconsistent from one type of allergy to the other?

Response 8: Thankyou for picking this up. We agree that this sentence is a little ambiguous, given our findings do reflect some of what has been reported in the literature. We have rephrased the abstract and conclusion to state:

“A range of foods in the pre-conception period were protective of any allergy, particularly saturated and unsaturated spreads, and both low and high fat dairy.”

Comment 9:Introduction/line40: incidence of food allergies in Australia also very high!??!? Please state and give references

Response 9: Thankyou, we agree with this comment and now include in the introduction:

“In Western countries, the prevalence of food allergy in children is around 4-7% but in up to 10% of infants has been reported in Australia (Loh W, 2018)”

Comment 10:“Former smokers”: for how long at minimum have these individuals been non-smokers

Response 10: There was no minimum time required to be a former smoker, but just not to have been continuing to smoke during pregnancy. The median duration that the former smokers had last smoked was a median 3 years ago, which we now state in the methods.

Comment 11:Saturated spread: give example

Response 11: We have noted ‘butter’.

Comment 12:Include abbreviation list or give full name when first mentioned (e.g. for SES)

Response 12: We no longer abbreviate and just report as “socioeconomic status”

Comment 13:Line 69: give citation if this first prospective study is already published

Response 13: Done

Comment 14:Line 91: and women who proceed with smoking during their pregnancy?! (acc. to table 1, this would be 15%)

Response 14: Yes, this is correct that 15% of women in this group were continuing to smoke during their pregnancy. We have indicated “continuing to smoke in their pregnancy” in the methods.

Comment 15:Line 94: if there is access, a link to the full FFQ would be good to include in the paper

Response 15: A link to the version used in this study has been included, but please note that a newer version is available.

Comment 16:Line 101: “was determined by their general practitioner” – how? By assessing history and examining symptoms, or doing tests?

Response 16: To identify asthma, we used the question “Has your doctor ever told you that your child has asthma?” in the ISAAC questionnaire. Asthma was then defined if the parents reported “Yes” to this question. In clinical practice,the GP askedabout a history of recurrent wheeze and/or cough responding to bronchodilator treatment, therefore the asthma diagnosis is based on clinical grounds. We now include this in the methods section.

Comment 17:Table 1: “Asthma status” – was this assessed at gestational week 18? Did those women have asthma also already before conception?

Response 17: Asthma status was assessed at the first antenatal visit (around 13 weeks’ gestation). We now include in the methods: “Maternal asthma status was determined by the response to the question by the midwife ‘Have you been told by a doctor that you have asthma?’ and ‘Have you used any asthma medications in the last year like salbutamol or a preventer?’, and women were categorised as non-asthmatic or asthmatic.”

Comment 18:Table 1: for higher clarity, please list “non asthmatic” before “asthmatic” and not before “asthma exacerbations” (like listed in line 85/86)

Response 18: We have ‘non-asthmatic’ in two places in Table 1 – i.e. in the ‘asthmatic status’ section and then in the ‘asthma exacerbations’ section. I have highlighted the second occasion in Table 1 so please decide if you would like to keep this in; we feel it makes sense to leave here, so the total percentage adds to 100%.

Comment 19:Table 1: please add info why only n=214 for head circumference – data for others not available (not measured or info not given by mother?

Response 19: Done

Comment 20:Line 129: “shared a common food group” is not clear;

Response 20: We have changed to “Various foods within the same food group, for example, tomatoes, red/yellow vegetables, leafy green vegetables, which are all ‘vegetables’, were not consistently positively or negatively related to allergy status in children.”

Comment 21:Line 129: “meats/legumes” as a common food group?

Response 21: See response to comment 20 above. This part of the sentence has been removed. 

Comment 22:Line 135: “Other fish” – explain or give example

Response 22: We have clarified in the results and discussion to “Other fish (e.g. canned fish and non-fried fish)”.

Comment 23:Fig. 1: what is Vegemite?

Response 23: We have included in the results “Salty type foods including vegemite (a yeast extract spread, similar to Marmite), tomato sauce, and processed meats, were associated with the development of eczema.”

Comment 24:Please define “juice”

Response 24: We have defined as 100% fruit juice.

Comment 25:Line 173: again, define “fish” clearly or give example

Response 25: As above, we have defined to canned fish and non-fried fish.

Comment 26:Line 204: typo “and a and high”

Response 26: Fixed.

Reviewer 2 Report

Summary:

Overall the paper is well written with a sound project design and statistical analysis. Some further attention to detail as recommended in the comments mentioned below would improve the article further. Particularly attention should be paid to the terminology used, demarcating and appraisal of studies that assess the role of diet during the preconception/pregnancy/lactation phases and further emphasis on the possible affect of recall bias.

Specific comments:

Introduction:

·         Line 44: Define what is meant by the “periconception” period upon first mention, as readers may not be familiar with the differences between the terms preconception, perinatal, antenatal etc – see later comments on the most suitable term to use.

·         Paragraph beginning Line 44: EAACI have recently published a position paper about dietary fatty acids and allergy outcomes across the lifecourse, which could be referenced here.

·         “In regions of low socioeconomic status, diet quality is poorer [17, 18], potentially contributing to 58 higher rates of allergy in these groups”: I understand the point you are making here, however this is explained a little too simplistically, given the multiplicity of factors that influence diet, sociodemographics and allergy rates – suggest it is reworded, or else the second part of the sentence needs some references for evidence.

Methods section:

·         Please elaborate on the difference between childhood wheeze and childhood asthma as there is considerable debate at which age asthma can be validly diagnosed in young children – this could either go in the methods section or discussion.

Results:

·         Line 119 : “The median BMI was in the overweight range, and more than one third were obese (Table 1).” Please clarify at which timepoint BMI as calculated – is this preconception or during pregnancy weight status? – if it is at 13 weeks gestation then this should be noted as a proxy measure of usual weight status.

·         It is unclear what the prevalence of each allergic condition was reported at each time point? A table/figure with this information is essential for context.

Discussion:

·         Line 155: “of whom 60% were either overweight or obese” - need to qualify this with “when measured at the end of the first trimester of pregnancy”.

·         Lines 171: “Typically, only the consumption of fish or omega 3 polyunsaturated fatty acids appears to be protective, with other reported food intake showing null, positive, or adverse outcomes.” Statement requires referencing.

·         Line 199: “higher pre-pregnancy”: you need to be consistent throughout the whole article with terminology. The term pre-conception is used in the results section and periconception in the title and introduction. Either use the term periconception/preconception or pre-pregnancy and define the time period from the outset. Given that the dietary data refers to a full 12 months prior to pregnancy, it is arguable that preconception or pre-pregnancy is the most appropriate term to use. “Periconception” should only be used if you are also including dietary data from early pregnancy.

·         The discussion mentions studies during different stages of pregnancy. One of the issues with this research topic is the temporality aspect - some studies include pregnancy data in the first/second/third trimester, others include lactation data and others include both. This issue should be mentioned. It would be useful to provide a very brief outline on the possible mechanisms that may influence allergic outcomes at each stage. i.e. what is the mechanism underlying the influence of preconception nutrition on allergic outcomes? Is it known whether preconception nutrition is more or less important than nutrition during pregnancy and lactation and why would this be?

·         Dietary recall of intake of allergenic foods around pregnancy and childhood is particularly problematic and a major limitation. Most studies use FFQs to collect data on the previous 1-3 months rather than 12 months. Suggest read and refer to paper by Van Zyl, Z et al. (2016). Also consider if negative dietary habits in early pregnancy affected by nausea may influence recall of preconception dietary habits. There is a nice paper by Crozier et al 2017 which outlines dietary changes in early pregnancy.

·         Other limitations to consider: Did you record intake of dietary supplements? E.g. vitamin D/omega 3? And how may this have affected results?

·         Role of reverse causation? i.e. those who have an atopic family history may deliberately exclude/include certain foods.

Author Response

Reviewer 2:

Summary:

Overall the paper is well written with a sound project design and statistical analysis. Some further attention to detail as recommended in the comments mentioned below would improve the article further. Particularly attention should be paid to the terminology used, demarcating and appraisal of studies that assess the role of diet during the preconception/pregnancy/lactation phases and further emphasis on the possible affect of recall bias.

Specific comments:

Introduction:

Comment 1: Line 44: Define what is meant by the “periconception” period upon first mention, as readers may not be familiar with the differences between the terms preconception, perinatal, antenatal etc – see later comments on the most suitable term to use.

Response 1: As per comment 9 below, we have changed to ‘pre-conception’.

Comment 2: Paragraph beginning Line 44: EAACI have recently published a position paper about dietary fatty acids and allergy outcomes across the lifecourse, which could be referenced here.

Response 2: Thankyou. This is a useful reference which we cite in the introduction and discussion.

Comment 3: “In regions of low socioeconomic status, diet quality is poorer [17, 18], potentially contributing to 58 higher rates of allergy in these groups”: I understand the point you are making here, however this is explained a little too simplistically, given the multiplicity of factors that influence diet, sociodemographics and allergy rates – suggest it is reworded, or else the second part of the sentence needs some references for evidence.

Response 3: Thankyou for this comment. There is no literature on the interaction between diet, sociodemographics and allergy. We have rephrased the second part of the sentence to state: and both low socioeconomic status (Almqvist C, 2005) and poor diet quality (Grieger JA, 2013; Venter C, 2019) associate with higher risk for allergy.

Methods section:

Comment 4: Please elaborate on the difference between childhood wheeze and childhood asthma as there is considerable debate at which age asthma can be validly diagnosed in young children – this could either go in the methods section or discussion.

Response 4: Thankyou, we agree that there is debate around this and that there is vogue in the international literature to use the term “preschool wheeze” rather than the asthma label, particularly in the younger age group. The ISAAC questionnaire offers two alternative quantitative measures of the frequency of wheezing to help determine differences in wheeze and asthma. For the purpose of this study, to identify asthma (and not wheeze as this was not an outcome), we used the question “Has your doctor ever told you that your child has asthma?”. Asthma was then defined if the parents reported “Yes” to this question. In clinical practice, the GP asked about a history of recurrent wheeze and/or cough responding to bronchodilator treatment, therefore the asthma diagnosis is based on clinical grounds.

Results:

Comment 5:Line 119 : “The median BMI was in the overweight range, and more than one third were obese (Table 1).” Please clarify at which timepoint BMI as calculated – is this preconception or during pregnancy weight status? – if it is at 13 weeks gestation then this should be noted as a proxy measure of usual weight status.

Comment 5: Thankyou, we have included that BMI was used a proxy measure of usual weight status.

Comment 6:It is unclear what the prevalence of each allergic condition was reported at each time point? A table/figure with this information is essential for context.

Response 6: Because this is a secondary analysis, we do not have data for each of the time-points indicated in the methods. We now state this in the methods, and as per comment 5 above, we have included child allergy status at 3 years in Table 1:

Allergy status at 3 years, n (%)

Any allergy / No allergy

188 (80%) / 46 (20%)

Asthma / No asthma

131 (56%) / 103 (44%)

Rhinitis / No rhinitis

93 (40%) / 141 (60%)

Eczema / No eczema

105 (45%) / 129 (55%)

Discussion:

Comment 7:Line 155: “of whom 60% were either overweight or obese” - need to qualify this with “when measured at the end of the first trimester of pregnancy”.

Response 7: Thankyou, we have included this.

Comment 8:Lines 171: “Typically, only the consumption of fish or omega 3 polyunsaturated fatty acids appears to be protective, with other reported food intake showing null, positive, or adverse outcomes.” Statement requires referencing.

Response 8: Done

Comment 9: Line 199: “higher pre-pregnancy”: you need to be consistent throughout the whole article with terminology. The term pre-conception is used in the results section and periconception in the title and introduction. Either use the term periconception/preconception or pre-pregnancy and define the time period from the outset. Given that the dietary data refers to a full 12 months prior to pregnancy, it is arguable that preconception or pre-pregnancy is the most appropriate term to use. “Periconception” should only be used if you are also including dietary data from early pregnancy.

Response 9: Thankyou for this comment. We only refer to dietary intakes prior to pregnancy so we have changed to pre-conception (title, and throughout the manuscript).

Comment 10: The discussion mentions studies during different stages of pregnancy. One of the issues with this research topic is the temporality aspect - some studies include pregnancy data in the first/second/third trimester, others include lactation data and others include both. This issue should be mentioned. It would be useful to provide a very brief outline on the possible mechanisms that may influence allergic outcomes at each stage. i.e. what is the mechanism underlying the influence of preconception nutrition on allergic outcomes? Is it known whether preconception nutrition is more or less important than nutrition during pregnancy and lactation and why would this be?

Response 10: Thankyou for this comment. We have now expanded on 2 paragraphs in the discussion with this information. Below in italics are the changes made to the previous paragraphs:

“Overall, the literature on maternal food intake and dietary patterns assessed during pregnancy and development of childhood allergy are inconsistent. Typically, only the consumption of fish or omega 3 polyunsaturated fatty acids appears to be protective [7-12], with other reported food intakes showing null, positive, or adverse outcomes [10-16]Moreover, there is uncertainty as to what time point during pregnancy, or whether before or straight after pregnancy, may have the biggest influence on allergy outcomes [16]. There is some data to suggest that perinatal nutrition may have an important function in programming immune function and allergy, in particular known immune-modulatory agents including long-chain omega-3 polyunsaturated fatty acids, antioxidant vitamins, vitamin D and folate [31], but the precise mechanisms operating before and during pregnancy that link maternal nutrition to offspring allergy development have not been clearly established.

We found that consumption of fish (i.e. canned fish and non-fried fish) may be protective for the development of child asthma and eczema, but was adversely associated with rhinitis. Only one recently published study was found assessing pre-pregnancy diet on allergies in children, finding no association between maternal fish intake in the 1 year prior to pregnancy and any type of child allergy [32]In a meta-analysis of maternal diet during pregnancy and infancy and risk of allergic or autoimmune disease, fish oil supplementation during pregnancy and breastfeeding was associated with a reduced risk of sensitization to food allergens but inconsistent results were found for the association with other allergic outcomes [16]. Observational studies however were inconclusive regarding consumption of fish or fatty acids on allergy outcomes [16]. While individual studies demonstrate a protective effect of maternal fish and omega 3 fatty acids intake on offspring allergy, the synthesized effect is less convincing. Further mechanistic studies are needed to identify the best time point for consuming fish and omega 3 fatty supplements, and the optimal intake to reduce the development of offspring allergy.”

Comment 11: Dietary recall of intake of allergenic foods around pregnancy and childhood is particularly problematic and a major limitation. Most studies use FFQs to collect data on the previous 1-3 months rather than 12 months. Suggest read and refer to paper by Van Zyl, Z et al. (2016). Also consider if negative dietary habits in early pregnancy affected by nausea may influence recall of preconception dietary habits. There is a nice paper by Crozier et al 2017 which outlines dietary changes in early pregnancy.

Response 11: Thankyou for this comment. We acknowledge that responses to any FFQ can be biased which we have indicated in the discussion. Although food allergy was not assessed in this cohort, we add to our current discussion on bias in FFQ by adding the following:

“Although the FFQ is widely used in dietary studies and has been validated in several populations [48-50], there are a number of biases that can be easily introduced into measuring diet including those associated with participant recall, compliance, and even intake of allergenic foods in pregnancy (van Zyl Z, 2016).

In regards to food changes during pregnancy, we are aware of the limited studies addressing this, typically showing minimal changes before and during pregnancy, like that of Crozier et al, and there is the possibility that nausea may affect food consumption and recall during pregnancy. Whilst there are several biases associated with food intake recall, we believe we have summarised those most pertinent to our study:

“Recall bias may have been particularly problematic since participants were asked to recount their pre-conception diet only after they had conceived. Second, we are limited regarding statements about the causal link between pre-conception diet and allergy status due to the observational nature of our study. For example, diet quality has consistently been associated with maternal BMI and smoking status, which are driven by socio-demographic factors. Untangling the complex interplay between these factors was something not within the scope of the present study.”

Comment 12:Other limitations to consider: Did you record intake of dietary supplements? E.g. vitamin D/omega 3? And how may this have affected results?

Response 12: We now include sentences on the problem of including confounders to LDA analyses. We appreciate this is a limitation but appropriate given the exploratory nature of the study.

The LDA analysis is a true multivariate method and does not really lend itself to confounding bias adjustment that we see in classical biostatistical models such as logistic regression. Thus, while LDA accounts for correlations among the individual dietary items when classifying allergy status, the impact of other maternal exposures such as body weight, medication and supplement use, smoking, breastfeeding, and even hygiene practices, cannot be fully elucidated.”

Comment 13:Role of reverse causation? i.e. those who have an atopic family history may deliberately exclude/include certain foods.

Response 13: Thankyou for this comment. We now include a paragraph on this in the discussion:

“Some of our results may be due to reverse causation. While it may be expected that women with a family history of allergy have poor quality diets, contributing to allergy in the child, we found that some healthier foods were associated with child allergy. This may be indicative of behaviour change bias, where individuals adopt more positive behaviours over time or even exclude certain foods based on allergy diagnosis. As the newly adopted dietary intake is assessed, this systematic misclassification may result in an apparent association of a healthier diet with the development of child allergy, which could readily be interpreted as a protective effect of a poor quality diet. While it is not possible to identify, test and model out this bias given limitations inherent in our study design, if present, such systematic misclassification would be expected to bias risk estimates towards the null.”